**www.cambridge.org/ext**

biocultural; culturally important species; anthropogenic Allee effect; culturomics

**Corresponding author:**
Richard J. Ladle;
Email: richardjamesladle@gmail.com

# Biocultural aspects of species extinctions

Richard J. Ladle[1,2,3] ⓘ, Fernanda Alves-Martins[1,2] ⓘ, Ana C.M. Malhado[1,2,3], Victoria Reyes-García[4,5,6] ⓘ, Franck Courchamp[7], Enrico Di Minin[8,9,10] ⓘ, Uri Roll[11], Ivan Jarić[7,12] and Ricardo A. Correia[8,9,13] ⓘ

[1]CIBIO-InBIO, Research Centre in Biodiversity and Genetic Resources, University of Porto, Vairão, Portugal; [2]BIOPOLIS Program in Genomics, Biodiversity and Land Planning, CIBIO, Vairão, Portugal; [3]Institute of Biological and Health Sciences, Federal University of Alagoas, Maceió, Brazil; [4]Institució Catalana de Recerca i Estudis Avançats, Barcelona, Spain; [5]Institut de Ciència i Tecnologia Ambientals, Universitat Autònoma de Barcelona, Cerdanyola del Vallès, Spain; [6]Departament d'Antropologia Social i Cultural, Universitat Autònoma de Barcelona, Cerdanyola del Vallès, Spain; [7]Ecologie Systématique Evolution, Université Paris-Saclay, CNRS, AgroParisTech, Gif-sur-Yvette, France; [8]Helsinki Lab of Interdisciplinary Conservation Science, Department of Geosciences and Geography, University of Helsinki, Helsinki, Finland; [9]Helsinki Institute of Sustainability Science (HELSUS), University of Helsinki, Helsinki, Finland; [10]School of Life Sciences, University of KwaZulu-Natal, Durban, South Africa; [11]Mitrani Department of Desert Ecology, The Jacob Blaustein Institutes for Desert Research, Ben-Gurion University of the Negev, MidreshetBen-Gurion, Israel; [12]Biology Centre of the Czech Academy of Sciences, Institute of Hydrobiology, České Budějovice, Czech Republic and [13]Biodiversity Unit, University of Turku, Turku, Finland

## Abstract

Predicting whether a species is likely to go extinct (or not) is one of the fundamental objectives of conservation biology, and extinction risk classifications have become an essential tool for conservation policy, planning and research. This sort of prediction is feasible because the extinction processes follow a familiar pattern of population decline, range collapse and fragmentation, and, finally, extirpation of sub-populations through a combination of genetic, demographic and environmental stochasticity. Though less well understood and rarely quantified, the way in which science and society respond to population decline, extirpation and species extinction can also have a profound influence, either negative or positive, on whether a species goes extinct. For example, species that are highly sought after by collectors and hobbyists can become more desirable and valuable as they become rarer, leading to increased demand and greater incentives for illegal trade – known as the anthropogenic Allee effect. Conversely, species that are strongly linked to cultural identity are more likely to benefit from sustainable management, high public support for conservation actions and fund-raising, and, by extension, may be partially safeguarded from extinction. More generally, human responses to impending extinctions are extremely complex, are highly dependent on cultural and socioeconomic context, and have typically been far less studied than the ecological and genetic aspects of extinction. Here, we identify and discuss biocultural aspects of extinction and outline how recent advances in our ability to measure and monitor cultural trends with big data are, despite their intrinsic limitations and biases, providing new opportunities for incorporating biocultural factors into extinction risk assessment.

## Impact statement

Human responses to impending extinctions are complex, are highly dependent on cultural and socioeconomic context, and have typically been far less studied than the ecological and genetic aspects of extinction. Specifically, the way in which science and societies respond to population decline, extirpation and species extinction can also have a profound influence, either positively or negatively, on whether a species goes extinct. For example, while some rare species suffer higher extinction risk the rarer they become, some charismatic species benefit from significantly higher conservation efforts and elevated levels of scientific research. A more comprehensive and nuanced understanding of which species will go extinct, and which will be "rescued" by conservation and stewardship efforts, requires an explicit interdisciplinary, biocultural approach to extinction that draws on expertise from the natural and social sciences, and dialog with holders of different knowledge systems and, in particular, with Indigenous Peoples and local communities. Ultimately, many currently threatened species will only go extinct if society allows it to occur, through a lack of motivation, knowledge, resources, or local conservation capacity.

## Introduction

Extinction is typically viewed as a logical end point of the process of population decline – the point on the graph where the population size curve meets the x-axis and terminates abruptly and

finally (Ladle and Jepson, 2008). Accordingly, the International Union for the Conservation of Nature (IUCN) defines a species as extinct if there is no reasonable doubt that the last individual has died (Hughes et al., 2021). Reasonable doubt in this context is the lack of evidence in the face of exhaustive surveys or extrapolation from historical observations (Solow, 2005). The intrinsic uncertainty about extinction led E. O. Wilson to characterize it as "the most obscure and local of all biological processes" (Wilson, 1992, p. 255). If scientists were to restrict their analysis to only well-documented extinctions, there would be a huge risk of underestimating current extinctions (Pimm et al., 2014). Thus, extinctions are often extrapolated onto unknown species, whose existence is inferred from species discovery curves, from biodiversity ratios, or from species–area relationships (Bebber et al., 2007; Chisholm et al., 2016; Kunin et al., 2018; García-Robledo et al., 2020).

Early conceptualizations of the extinction process strongly emphasized the role of population decline and the effects of small population size on population viability (Caughley, 1994). The former process is the result of deterministic factors such as habitat loss, degradation and overexploitation, ultimately leading to small fragmented populations that are highly susceptible to stochastic factors (Lande, 1998). Accordingly, extinction risk assessment schemes such as the IUCN Red List emphasize the rate of population decline, distributional range, and the size and fragmentation of extant populations as important factors that influence long-term prospects of a species' survival (https://www.iucnredlist.org/assessment/process). Likewise, these important insights led directly to the development of the concept of minimum viable populations and sophisticated tools for performing population viability analysis (Ladle, 2009; Traill et al., 2010; Flather et al., 2011).

In summary, extinction is typically understood to mean the death of the last individual of a species and is a consequence of the process of population decline and the negative consequences of small population size. However, contemporary extinction is almost never a purely biological process (Ladle and Jepson, 2008). Rather, contemporary extinction is almost always a consequence of the interaction of cultural and biological phenomena, that is a *biocultural* process. Indeed, cultural practices now influence almost every aspect of the extinction process – how we perceive it, measure it and act upon it – and, critically, whether a threatened species actually goes extinct (or stays extinct), echoing calls for the adoption of biocultural approaches to conservation (Garibaldi and Turner, 2004; Gavin et al., 2015; Bridgewater and Rotherham, 2019). We contend that there is a need for a broader conceptualization and exploration of species extinctions as a biocultural process. Here, we outline some of the key biocultural aspects of extinction, including taxonomic change, human impacts on population decline, and how the behavior of the global conservation movement plays a key role in contemporary extinction dynamics.

## Taxonomy and extinction

Although extinction is sometimes used to describe the extirpation of populations, geographic variants or subspecies (often called "local extinction"), the term is more commonly applied to the loss of all populations of an officially recognized species (Ladle and Jepson, 2008). Extinction statistics are therefore highly sensitive to changes in taxonomic practice, especially changes in normative use of species concepts and species delimitation criteria (Zachos, 2016). Such changes have become increasingly prevalent, partly as a consequence of advances in molecular taxonomy, leading to

significant recent increases (and sometimes decreases) in the number of recognized species in many taxa (Garnett and Christidis, 2017). Every taxonomic decision to "split" a species into two or more species or to "lump" two or more species into a single species necessarily has consequences for the extinction risk of each newly defined species (Mace, 2004). Splitting means that each newly recognized species will have a smaller population size and/or geographic range, potentially increasing the threat level. Such increasing threat may potentially be compensated by the increased conservation attention afforded to a fully recognized species rather than a subspecies or regional variant. It could also reduce the average societal attention given to each newly designated species (Ladle et al., 2019). Moreover, estimates of the numbers of unknown species (sometimes referred to as the Linnean Shortfall; Hortal et al., 2015) are highly sensitive to the number of documented species, with knock-on effects for estimates of global extinction rates (Stropp et al., 2022).

## Cultural push-factors

The first scientists to study extinction at the end of the nineteenth century had enormous difficulties attributing the decline and eventual loss of a species to human actions (Ladle and Jepson, 2008). For example, despite clear evidence of overhunting (Bengtson, 1984), James Orton described his confusion about the causes of the extinction of the Great Auk (*Pinguinus impennis*) as follows: "The upheaval or subsidence of strata, the encroachments of other animals, and climatal revolutions—by which of these great causes of extinction now slowly but incessantly at work in the organic world, the Great Auk departed this life, we cannot say" (Orton, 1869, p. 540). This reluctance to attribute human causes to species extinction continued well into the twentieth century (Ladle and Jepson, 2011). In contrast, modern current conceptualizations of the factors driving population declines foreground the indirect and direct role of human actions and how they are shaped by sociocultural practices and beliefs (Lande, 1998; Díaz et al., 2019).

Although extinction can occur in the absence of human influence (De Vos et al., 2015), the vast majority of contemporary extinctions are ultimately or proximately connected to human action (Ceballos et al., 2015; Díaz et al., 2019, 2015), and are underpinned by societal values and behaviors. Ultimate causes include human population growth (McKee et al., 2004), the seemingly universal desire to accumulate surplus capital (McBrien, 2016), the political need for economic growth (Spash and Smith, 2019) and the grinding hardship of rural poverty that forces individuals into a reliance on natural resource exploitation (Adams et al., 2004). These factors, in turn, drive the proximate causes of population decline, the most significant of which are habitat loss, fragmentation and transformation (Maxwell et al., 2016; Powers and Jetz, 2019), climate change (Thomas et al., 2004; Cahill et al., 2013), biological invasions (Clavero and García-Berthou, 2005) and overexploitation (Bennett et al., 2002; Maxwell et al., 2016).

While habitat loss, climate change and biological invasions can be considered as by-products of other human activities such as agriculture, trade, transport and recreation, population decline due to overexploitation of species is a direct consequence of human cultural practices and values. This is clearly illustrated by the anthropogenic Allee effect, the idea that human predisposition to place an exaggerated economic value on rarity drives the disproportionate exploitation of rare species, causing them to become rarer and therefore even more desirable (Courchamp et al., 2006;

Palazy et al., 2012a; Tournant et al., 2012). Even when a species becomes so rare that it cannot, alone, support livelihoods, opportunistic exploitation, while targeting more common species, will ensure that population decline continues (Branch et al., 2013). For example, Chinese bahaba (*Bahaba taipingensis*) is a highly sought-after fish for use in traditional Chinese medicine, but fishers seeking this species must make their living off other species because only a few are caught each year (Sadovy and Cheung, 2003). The anthropogenic Allee effect, an explicitly biocultural model of extinction risk, is particularly applicable to "collectable" exotic species (Siriwat et al., 2019) or their products, as in the example of traditional Chinese medicine highlighted above (but see Mateo-Martín et al., 2023). More generally, it illustrates the complex ways in which cultural practices and beliefs can become entangled in the process of population decline and extinction. In this case, human perceptions of rarity and economic dynamics interact with the population trends of the exploited species to create an "extinction vortex" (Courchamp et al., 2006).

The ultimate and proximate drivers of population declines of species are typically considered to be insufficient to cause the actual extinction of a species (Lande, 1998). Instead, populations become so small and fragmented that they become subject to a range of stochastic processes (genetic drift, demographic and environmental stochasticity, natural catastrophes) that ultimately lead to the death of the last individual (extinction) (Lande, 1998). As described above, in a few cases the increasing rarity value of these last individuals vastly inflates their economic value and therefore incentivizes their capture or elimination (Courchamp et al., 2006; Hall et al., 2008). In species without economic value and without sufficient human intervention (see below), remnant populations eventually succumb to one of the many risk factors associated with small populations, as highlighted by individual animals that become famous for being the "last" of their species (Nicholls, 2012; Jarić et al., 2023).

## Cultural push-back factors

While cultural "push-factors" for extinction are generally well known and quantified, far less attention has been given to the role of humans in delaying or preventing extinction ("push-back" factors). Since the emergence of the global conservation movement in the late nineteenth century (Soulé, 1985), conservationists have increasingly monitored threatened populations and, when deemed necessary, intervened in aiming to halt or slow the extinction process (Hoffmann et al., 2015; Bolam et al., 2021).

The increasing capacity of the global conservation community to identify species at risk of extinction and to take action to mitigate this risk highlights the key role that humans now play in the extinction process. In other words, species become extinct (or avoid impending extinction) due to the interplay between human-mediated biological processes (range collapse, population decline, small populations) and human capacity to monitor and successfully intervene in this process (Ladle and Jepson, 2008). There is a global safety net provided by the conservation movement as represented by government bodies and various international and national non-governmental conservation organizations (NGOs). In a similar way that extinction threats vary geographically, the motivation, capacity, resources and effectiveness of conservation also vary immensely by country and region (Waldron et al., 2013). Sometimes extinction threats and capacity to deal with those threats align, but frequently there is a mismatch, with geographic areas hosting a high frequency of threatened species mainly located in the "Global South" (Schipper et al., 2008) while the capacity of the global conservation movement to act is often more concentrated in the "Global North" (Balmford et al., 2003). While this is generally true, there are many exceptions, and further research into this area is needed.

Although conservation capacity is clearly central to the probability and time-scale of extinction for many species, it is a very poorly defined concept, hampering measurement and mapping efforts. The broadest definition of conservation capacity in relation to species extinctions includes at least three dimensions (Table 1): i) willingness and motivation to act; ii) knowledge to design effective interventions; and iii) institutional, technical and economic resources to implement effective interventions. The interactions of these three dimensions will largely determine whether a species is identified as being at risk of extinction, whether efforts are made to reduce the risk of extinction and whether those efforts are successful in the short and long term.

### *Cultural willingness and motivation to prevent extinction*

It is self-evident that societies vary enormously in their willingness to allocate resources to conserve different species depending upon their values and valuation of nature (Díaz et al., 2015). Among the conservation community, charismatic and culturally iconic species of vertebrates (mainly mammals and birds) are prioritized for conservation funding and action over equally threatened but culturally less visible species (Davies et al., 2018; Mammola et al.,

**Table 1.** Main dimensions of conservation capacity and some of the methods of measurement (see text for details)

| Dimension | Definition | Example methods | References |
|---|---|---|---|
| Willingness and motivation to act | Societies vary in their desire to prevent different species from going extinct, largely depending on the societies' cultural values and on the species' cultural characteristics (public awareness, interest, sentiment, etc.) | Culturomic analysis | Millard et al. (2020) |
| | | Social surveys | Samojlik et al. (2023) |
| Knowledge | The amount of scientific and/or local knowledge of species that is relevant to their conservation, management and stewardship varies due to a wide range of cultural and historic factors | Bibliometrics | dos Santos et al. (2020) |
| | | Expert assessment | Pearce-Higgins et al. (2017) |
| | | Reports from Indigenous and local knowledge systems | Ziembicki et al. (2013) |
| Institutional, technical and economic resources | The capacity of local actors and conservation organizations (NGOs, governmental bodies, international institutions and private organizations) to fund and implement successful conservation interventions varies geographically in relation to complex socioeconomic, political and historic factors | Desk-based analysis of institutional capacity | Malhado et al. (2020) |
| | | Social surveys | Fu and Shumate (2020) |

2020). A recent analysis of the internet salience of 36,873 vertebrate taxa revealed that search interest was higher for more threatened mammal and bird species than it was for fish, reptiles and amphibians (Davies et al., 2018). Similarly, Kim et al. (2014) examined web search data for 246 threatened species in Korea and found that the interest for mammals, birds, amphibians and reptiles were 10 times higher than those for other taxa. This bias toward vertebrates also has a geographical component, with temperate species receiving more conservation attention than those in the tropics (Titley et al., 2017). Although plants generally receive less conservation attention than vertebrates, they are also strongly influenced by cultural perceptions (Adamo et al., 2021), with a recent study indicating that species with attractive flowers received more funding, irrespective of extinction risk (Adamo et al., 2022). Similarly, fungi conservation is significantly biased toward macrofungi since these are most easily observed and include many edible taxa (Gonçalves et al., 2021). It should be noted that even charismatic vertebrate species may still be lacking adequate resources to prevent continued population decline (Di Minin et al., 2015; Courchamp et al., 2018).

Culturally prominent species that generate high public interest and positive sentiment are more likely to be the target of conservation actions for two main reasons. Firstly, it is easier to mobilize support and resources through campaigns and other fund-raising actions when a species already has a high public profile (Thomas-Walters and J Raihani, 2017). Secondly, societal preferences also extend into scientific research, with researchers across the world preferentially collecting data on larger, more charismatic taxa independent of the threat status (Caro, 2007; Troudet et al., 2017). Conversely, many species receive little to no attention and are likely to suffer from a process of societal extinction – the decline of collective attention and memory of an extinct or threatened, extant species (Jarić et al., 2022). The process of societal extinction of species is linked to that of biological extinctions, as it is likely to result in decreased support for conservation action, ultimately affecting negatively the outcome of such efforts.

Until recently it was challenging to quantify the level of public awareness, interest and sentiment about threatened species at a national, regional and global scale because this required the use of time-consuming and costly social surveys. However, the recent development of "conservation culturomics" (Di Minin et al., 2015; Ladle et al., 2016; Correia et al., 2021), the analysis of digital data generated by people to provide novel insights on human–nature interactions, allows the evaluation of multiple aspects of societal preferences for species and higher taxa. For example, Ladle et al. (2019) found that the salience of bird species on the global internet was strongly correlated with species that have wide geographic ranges that overlap with technologically advanced societies, that are phenotypically conspicuous and, critically, that have direct interactions with humans (e.g., hunting, pet keeping, etc.). A related study based on Wikipedia page views for all extant species of birds found that farmed species and species in the pet bird trade were particularly prominent over multiple language editions (Mittermeier et al., 2021). These examples demonstrate how culturomic metrics can be used to capture and quantify different aspects of human interest in nature at scales that are beyond the reach of standard social surveys (see also Fink et al., 2020; Falk and Hagsten, 2022; Johnson et al., 2023).

Big data approaches such as culturomics have enormous potential but also many limitations related to scale and coverage (reviewed in Correia et al., 2021, Di Minin et al., 2021). For example, many Indigenous Peoples (and many other socially and economically marginalized groups) have limited access to the global

internet, and understanding their interactions, attitudes and sentiment toward local species is also critical for effective conservation, but frequently ignored in conservation management (Zanotti and Knowles, 2020). Recognizing Indigenous Peoples and local communities' rights and agency in conservation management (Reyes-García et al., 2022) is of critical importance, both because much of the world's biodiversity now exists in landscapes and seascapes traditionally owned, managed, used and/or occupied by Indigenous Peoples (Garnett et al., 2018) or by local communities (Brondizio and Le, 2016) and because such a strategy might ultimately improve conservation outcomes (Büscher and Fletcher, 2019). Rates of biodiversity decline are slower in such areas than elsewhere, including protected areas (Garnett et al., 2018; Fa et al., 2020; O'Bryan et al., 2021). Reyes-García et al. (2023) recently developed a framework around the concept of culturally important species that could be used to integrate different nature values into the management of threatened species. Such species predominated among areas where Indigenous Peoples live and, critically, include a high proportion of species that the IUCN classify as Data Deficient. Species in their study were more likely to be culturally than biologically threatened, especially those associated with Indigenous Peoples due to the high levels of cultural loss they have experienced.

### Conservation-relevant knowledge of species

It has long been recognized that there are large and persistent taxonomic biases in which species have been researched and, consequently, the volume and quality of scientific knowledge about different species (Clark and May, 2002; Fleming and Bateman, 2016). Such variations potentially have a significant impact on the capacity of societies to prevent species from going extinct. For many species, to be "saved" from extinction there should be sufficient biological, ecological and cultural knowledge of the species and its habitat to support the design and implementation of appropriate conservation interventions (Murray et al., 2015; Cooke et al., 2017). For other species, habitat and site-based conservation may be sufficient to prevent extinction and population decline. Moreover, more knowledge of a species or higher taxon does not always lead to better conservation interventions or swifter responses when a species is threatened, but, all things being equal, a well-studied species or group is more likely to be the subject of effective conservation actions than a poorly known counterpart. It should be noted that it is not only published scientific knowledge that is potentially important but also the practical and contextual knowledge of researchers (and other stakeholders) about the species in question. The greater the research effort, the greater number of people with such knowledge that can be mobilized to facilitate conservation efforts.

The causes of these biases are relatively well understood (Jarić et al., 2019). For example, scientists tend to study species within the country where they work due to a combination of funding priorities, cost and convenience. It follows that countries with low scientific capacity typically have fewer qualified conservation scientists and ecologists and less resources available for research, leading to geographical biases in conservation research effort (Meyer et al., 2015). Species also vary in their "researchability" – any characteristic of the species that potentially increases the costs of data collection or which impedes or reduces the feasibility of a research project (da Silva et al., 2020; dos Santos et al., 2020). For field-based conservation research, this could include characteristics that make it harder to observe a species, such as small body size, habitat characteristics and accessibility, nocturnal activity patterns,

elusiveness or cryptic coloration. Researchability could also be correlated with "conservability," if the traits that make a species more challenging to study overlap with those that make it more difficult to implement conservation interventions. Moreover, when species are challenging to study they become less desirable targets for researchers whose chances of career advancement may depend on their publication record or the completion of a high-level research dissertation (Caro, 2007).

As with human interest in species (see above), the last decade has seen great advances in our capacity to quantify taxonomic biases in research at scale through the analysis of bibliometric databases such as Scopus, Web of Knowledge or Google Scholar. For example, a recent regional-scale bibliometric analysis of Australian birds demonstrated significantly more publications on species with larger body sizes, larger ranges, higher relative abundance and presence in urban environments (Yarwood et al., 2019). A similar study on all extant species of mammals found that research volume was strongly associated with the scientific capacity within the range of species, high body mass and whether the species was non-native, with a very weak effect of conservation threat status (dos Santos et al., 2020).

Additionally, it should be noted that there can be a mismatch between the "researchability" of a species, as determined by scientists, and its cultural relevance, as defined by local criteria (Crane et al., 2016). Reyes-García et al. (2023) found that culturally important species had a much higher proportion of Data Deficient species than the full set of IUCN species, most likely resulting in an underestimation of their biological threat, as species categorized as Data Deficient by the IUCN seem to be more threatened than data-sufficient species (Borgelt et al., 2022). The data gap underscores that cultural considerations remain disregarded in much current biological research (Bridgewater and Rotherham, 2019) despite the fact that Indigenous and local knowledge has long been deemed as essential to setting realistic and effective biodiversity targets (Berkes et al., 2000; Brondízio et al., 2021; Reyes-García et al., 2022).

### Institutional, technical and economic capacity to intervene

Even when there is good scientific knowledge about a threatened species and strong public support for conservation action, weak institutional capacity means that interventions may be poorly planned and executed or not even implemented. In this context, institutional capacity normally refers to governmental departments, conservation NGOs and other civil society groups, and occasionally private sector organizations. Measuring such capacity is highly challenging, and there have been very few systematic analyses of conservation organizations at national, regional or global scales (Brockington et al., 2018). To our knowledge, there have not yet been attempts to evaluate institutional conservation capacity at the level of species or geographic regions (e.g., countries), though such quantifications could play a major role in determining the number, type and quality of interventions in the face of endangerment (Ladle and Jepson, 2008). Such a lack is partly attributable to the difficulties of collecting data on diverse conservation actors (Malhado et al., 2020), and partly due to the complexity of factors that contribute to institutional capacity, severely limiting the potential to develop robust metrics.

Over the last decades, there has been an institutionalization of co-management and bottom-up approaches to conservation (e.g., Indigenous and Community Conserved Areas, community monitoring). For example, community-based monitoring is increasingly proposed to improve scientific understanding of biodiversity status and trends, or local uses of plants and animals, among other processes (Danielsen et al., 2021). Understanding the role of such initiatives in curving extinction processes also requires monitoring.

### How many species have been "saved" from extinction?

Estimating the number of species that would have gone extinct if conservationists had not intervened is, by definition, exceedingly challenging, and the fact remains that the number of species currently threatened with extinction is unprecedented in human history (Ceballos et al., 2010; Díaz et al., 2019). Well-known examples of "near extinction" events, such as the black-footed ferret (Dobson and Lyles, 2000) or the Chatham Island Black Robin (von Seth et al., 2022), are indisputable. However, the imminent demise of the rescued population is often less clear-cut and there have been few large-scale estimations, mainly restricted to birds and mammals (Table 2). For example, Bolam et al. (2021) estimated the number of species "saved" by canvassing the opinion of experts; their estimate that bird and mammal extinction rates would have been 2.9–4.2 times greater without conservation action is almost certainly an underestimate given that only clear cases were considered. There are many more situations where, had conservation not intervened earlier in the extinction process (i.e., before a species becomes Critically Endangered), a species would arguably have gone extinct. Moreover, many species have avoided extinction due to actions aimed at conserving sites, habitats and ecosystems. This category of "saved" species is even more difficult to quantify since they include many lesser-known taxa, some of which may be undescribed (Hortal et al., 2015).

A closely related issue to species saved from extinction is the number of species that would eventually go extinct without continued conservation investment. Such "conservation-dependent" species could justifiably include, among others, species whose i) populations are periodically augmented with captive-bred individuals; ii) populations are not declining due to continued management efforts such as anti-poaching measures and surveillance, genetic management, control of invasive species, supplemental feeding, etc.; and iii) last remaining individuals exist only in captivity. Taking the latter group of species as an example, there are 85 species currently classified as Extinct in the Wild (i.e., only ex-situ populations remain), and some of these species have persisted in captivity for over 70 years (Smith et al., 2023). Other forms of conservation dependence are poorly quantified but potentially represent a significant limitation to

**Table 2.** Estimated number of bird and mammal species that would have gone extinct without direct human intervention

| Taxon | Number of species | Timeframe | References |
|---|---|---|---|
| Birds | 16 | 1994–2004 | Butchart et al. (2006) |
| | 21 to 32 | 1993–2020 | Bolam et al. (2021) |
| | 9 to 18 | 2010–2020 | Bolam et al. (2021) |
| Mammals (all) | 7 to 16 | 1993–2020 | Bolam et al. (2021) |
| | 2 to 7 | 2010–2020 | Bolam et al. (2021) |
| Mammals (Ungulates) | 6 | 1996–2008 | Hoffmann et al. (2015) |

future conservation actions, given the limited resources available for new initiatives.

## Extinction risk forecasting using push and push-back factors

As quantitative assessments of extinction risk, Red Lists are a crucial knowledge product and underpin much conservation law and policy (Rodrigues et al., 2006; Hoffmann et al., 2008). Red List categorizations are used, among other things, to i) support conservation decisions at, and across, multiple governance levels; ii) guide strategy and investments in species conservation; and iii) inform progress toward targets of international agreements. Perhaps more importantly, Red Lists have translated the key conservation value of avoiding anthropogenic extinctions into a governance tool that has helped produce global norms governing relations between society, economy and the nonhuman world (Jepson et al., 2011).

IUCN Red Lists assign species to extinction threat categories based on five quantitative criteria: i) population vulnerability; ii) population size reduction; iii) geographic range; iv) population size; and v) population viability analysis. These categories have a population ecology/life history focus, yet, as argued above, extinction (and its avoidance) is a biocultural phenomenon. Before and after a taxon is assigned to a Red List category it is subject to cultural forces that determine the success (or otherwise) of conservation actions (Ladle and Jepson, 2008). We would argue that a species well known to science is, *ceteris paribus*, less likely to be at risk of extinction compared with a lesser-known species in the same threat category because the public and institutions will mobilize more effectively to save it. Exceptions may include species that are highly sought after as pets, trophies, food or fashion accessories (Gault et al., 2008; Palazy et al., 2012b; Leclerc et al., 2015) that may suffer more intense exploitation than less-desirable species (Courchamp et al., 2006). Moreover, IUCN species lists do not explicitly include the importance of species for local cultures (Reyes-García et al., 2023), a factor that could also play a vital role in the success of any proposed conservation intervention. In short, IUCN Red List categories currently omit a range of non-biological factors that may be critical in determining whether a species will be "saved" from extinction (or not).

Creating a systematic and comprehensive system of assessing public interest and local cultural importance of species could, along with information of scientific knowledge of species, provide interesting complementary information to support and add nuance to extinction categorizations (Figure 1). Specifically, information from Indigenous and local knowledge systems or from macroscale

cultural analysis (e.g., culturomics) could potentially be used to i) identify threatened species assemblages and geographic regions (or parts of regions) where the potential for rallying support may be weaker and where greater investment may be required to improve the conservation status of a taxon (Ladle et al., 2016); ii) further support the use of Red Lists in business and investment decisions (Bennun et al., 2018), making material the reputational risks associated with activities that impact publicly visible globally threatened species; iii) enhance the quality of conservation actions by increasing the recognition of Indigenous Peoples and local communities' knowledge, values and rights (Reyes-García et al., 2022); and iv) provide complementary information to support and prompt innovative actions to reduce extinction risk. For example, "digital interventions" can raise the public profile of threatened species and more effectively link communities of interest with specific taxa.

The above suggestions come with the caveat that cultural evaluation based on big data approaches such as culturomics has many limitations and biases (Troumbis and Iosifidis, 2020; Correia et al., 2021), and much research is still needed to develop robust, well-validated metrics that can be used with confidence for conservation planning and assessment. Furthermore, as cultural metrics are eventually integrated into extinction risk assessment, it is inevitable that different threatened species might benefit or lose out depending on how conservation organizations choose to use this information; for example, deciding on the balance between funding the conservation of well-known species versus promoting the conservation of species deemed to have little or no cultural importance.

All things being equal, in areas where conservation capacity is low, i) species are less likely to be "saved" from extinction due to a lack of scientific knowledge, resources and effective conservation interventions; ii) threatened species may be less effectively monitored (Fisher et al., 2011) leading to incomplete knowledge of species distributions/population status and slow or absent conservation responses; and iii) technological interventions such as captive breeding, reintroductions and translocations are less likely to be implemented or successful. However, the willingness and capacity of institutions to act to prevent a species from imminent extinction are not straightforward to evaluate and partially depend on the cultural characteristics of the threatened species, with far less effort expended on the conservation of non-charismatic species. For example, Bellon (2019) found that species popularity (higher internet salience) had a greater effect than federal priority ranking for the funding of threatened species by various US federal agencies under the Endangered Species Act.

## Conclusions

Species extinction is a complex phenomenon that involves both biological and social factors. Understanding and addressing these factors is crucial for the conservation of biodiversity. Most, if not all, species currently in danger of imminent extinction are in that state due to the direct and/or indirect impacts of humans on the environment. Moreover, whether these species actually go extinct will largely depend on the willingness to act and the technical capacity of local, national and international conservation organizations, along with the support of local communities and other stakeholders. In contrast to the assessment of biological and ecological risk factors, our understanding and quantification of the cultural and political vulnerability of species is at an early stage of development. A more comprehensive understanding of which species will

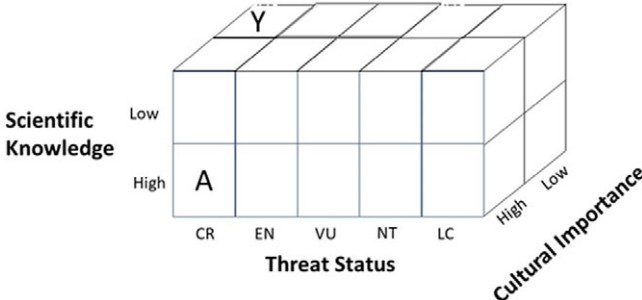

**Figure 1.** Enhanced extinction assessment by combining the IUCN Red List with metrics of scientific knowledge and cultural salience. Species in block Y are more likely to go extinct compared to block A because, in addition to biological drivers of extinction (e.g. small population size, small range size, declining populations, etc.), species in this category have low cultural salience, reducing willingness/motivation to act, and low scientific knowledge, reducing ability to mount effective conservation interventions

go extinct and which will be "rescued" by conservation and stewardship efforts will require an explicit interdisciplinary, biocultural approach to extinction that draws on expertise from the social sciences, and dialog with holders of other knowledge systems and, in particular, with Indigenous Peoples and local communities. For many currently threatened species, extinction will only occur if society allows it to occur, through a lack of motivation, knowledge, resources, or local conservation capacity.

**Open peer review.** To view the open peer review materials for this article, please visit http://doi.org/10.1017/ext.2023.20.

**Author contribution.** All authors contributed to the conceptualization, writing the original draft, and reviewing and editing of the draft. Writing was coordinated by R.J.L.

**Financial support.** R.J.L., F.A-M. and A.C.C.M. are supported by the European Union's Horizon 2020 research and innovation programme (grant agreement #854248). V.R-G. is supported by the European Research Council under an ERC Consolidator Grant (FP7–771056-LICCI). This research contributes to the "María de Maeztu Unit of Excellence" (CEX2019–000940-M). E.D.M. thanks the European Research Council (ERC) for funding under the European Union's Horizon 2020 research and innovation program (grant agreement 802,933). R.A.C. acknowledges funding from the Research Council of Finland (Grant agreement #348352) and the KONE Foundation (Grant agreement #202101976).

**Competing interest.** The authors declare none.

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
