## [Reviewer Report]

In this manuscript, the authors develop the theme of the correlation between the extinction of a taxa and the socio-cultural aspects that ensue or that may lead, in some cases, to the mitigation of the risk of this occurring; or, in other cases, the aggravation of the status of a species. Perhaps this second aspect is somewhat less investigated, but that may just be my impression.

The introductory part defining extinctions seemed a bit long-winded to me, it seems to skirt around the point, without sinking in; it is the only paragraph where I got this impression, in fact, in general the article is very interesting and well-written and does not bore despite the large amount of bibliography cited.

Being a kind of review, the article has no results to discuss, which is why I would have expected references to be given in the tables, perhaps not all of them, but at least the most significant articles (if I have not understood anything, ignore this suggestion). Reading the abstract of the manuscript, I would have expected to find some references to traditional Chinese medicine. Rhino horn is perhaps one of its best-known global effects, but the genus Dendrobium (Orchidaceae) is also not healthy due to its use in traditional medicine, and numerous other examples exist. I would also have expected to find more practical operational indications on how to integrate socio-cultural aspects with the risk of extinction.

In the first paragraph ‘Cultural willingness and motivation to prevent extinction’, it would be nice to add some indication of the conservation of macromycetes, which are often completely ignored. I recommend the article by Gonçalves et al. (2021).

In line 208, you talk about indigenous peoples and the difficulty of tracking their interactions, attitudes, and sentiment towards local species through the Internet. But it seems to me that there is a problem of scale in this kind of statement: it is obvious that through the web I can extract information that is valid for contexts that i) have access to the net ii) concern regional or global scale. Or else the message of the sentence is not very clear. At line 242 it occurred to me that there are places that are home to such excellence in nature that they attract research teams from all over the world (I was thinking of Kenya, Tanzania, Costa Rica, …)

Minor remarks:

- Keywords in the first page are different from those at page 3. I prefer the latter.

- R144: Mitigate in place of ameliorate? Sounds it better?

- R148: Movents

- R154-155: Global South and Global North should be in quotes

- R180: shown instead of showing

- R189: You can also cite Adamo et al. (2021) for plants

- R199: there is a problem with the 2019 in Ladle …

- R260: invasiveness looks like a very negative term. It should be used for alien species only. Pioneer species are not invasive.

- R262: cultural relevance instead of importance

- R298-312: several double )) and ;;

- R328: the list starts from ii) instead of i)

- R347: Bolham reference to fix

- Table2: Ungulates are mammals!

- R364: iii) instead of ii)

- R364: in the paragraph there is a weird mismatch of brackets.

-

Gonçalves, Susana C., Haelewaters, Danny, Furci, Giuliana, Mueller, Gregory M., 2021. Include all fungi in biodiversity goals. Science 373 (6553), 403-403.

Adamo, Martino, Chialva, Matteo, Calevo, Jacopo, Bertoni, Filippo, Dixon, Kingsley, Mammola, Stefano, 2021. Plant scientists’ research attention is skewed towards colourful, conspicuous and broadly distributed flowers. Nat. Plants 7 (5), 574–578.

---

## [Reviewer Report]

This paper provides an overview on how science and societies’ responses to the threat of extinction can alter the risk of a species going extinct. In doing so the paper also highlighting emerging methods for investigating cultural trends that are one of the factors determining these responses. The paper was novel in reviewing the information that is available on this understudied topic and highlighting how more information can be gathered and incorporated into existing estimates of extinction risk. Finally, this paper serves as a critique of current practices which do not explicitly incorporate social information when thinking about conservation. The issues raised in this paper should be more widely understood by people working in conservation and this paper has a lot to offer. Altogether this is a worthwhile contribution to the literature on the biocultural aspects of extinction.

However, the paper would be improved if it made clearer some of the limitations in the body of work that is being reviewed and possible limitations with the methods being advocated. Even if the new methods are very effective at understanding social trends, it was not clear to what extent social trends determinant of societal response to extinction. If there would still be a lot of uncertainty about societal response even with an understanding of trends, that should be clear in the text. Being very clear about limitations with these biocultural methods should be done if the authors are going to advocate for their widespread adoption via inclusion in metrics within the Red List.

It should also probably also be made clearer that more widespread adoption of biocultural metrics is potentially only the beginning of the conversation about how this information should be used. For example, focusing on species deemed culturally important may conflict with spending resources on improving perceptions of a species that are at risk of societal extinction. The authors do provide different ways biocultural information may be used, but there is no suggestion that different endangered species might benefit or lose out depending on how conservation organizations choose to use this information.

P6 L164-Table 1. Consider providing some other examples of assessment methods.

P7 L190-Briefly explain what “societal extinction” means.

P7 L191-Is there empirical evidence that societal and biological extinction would be “tightly linked”? For example, strategies such as conserving umbrella species could allow species that are not well known to still be biologically protected. Either provide a source or weaken the claim somewhat.

P8 L208- It is not only Indigenous People that have limited access to the global internet and whose interactions, attitudes and sentiments might affect conservation outcomes. Continue to highlight the important role of Indigenous communities, but also acknowledge that big data approaches could have limitations in other types of marginalized communities.

P8 L211-Acknowledging local people’s rights and agency in conservation management is critically important from a conservation perspective not because most biodiversity is on land owned, managed, or used by indigenous people, but presumably because acknowledging their rights and agency improves conservation outcomes. This is implied in the broader text, just reword the sentence.

P8 L217- “…(CIS) could contribute…”, does this “could” refer to CIS contributing if the method were more widely adopted, or referring to the benefits of the method not yet being completely clear?

P8 L221- If many of the CIS were data deficient, do we know they were “more likely to be culturally than biologically threatened,” or are we uncertain how biologically threatened CIS are?

P9 L266-Is this data gap only driven by researchability, as the paragraph implies, or might it also be driven by culturally important species being in countries with lower scientific capacity? It would seem the ease with which the gap might be filled would depend heavily on the driver behind it.

P9 L269-Including some older citations showing indigenous and local knowledge has been deemed essential to setting realistic biodiversity targets would help illustrate that this is a problem that has been known about but not acted upon. The very recent papers cited give the impression there may not have been the time for the benefit of any adjusted priorities to be seen, rather than implying that local knowledge is still being ignored.

P9 L270- On the sub section Institutional, technical and economic capacity to intervene: Are there general metrics of institutional capacity at the national level that could help understand the institutional, technical and economics capacity to intervene in the absence of conservation specific metrics?

P10 L298- The paragraph starting at this line is mostly talking about threats that are not acknowledged in the current red list, but at the end seems to start talking about how species that have local value should be highlighted, perhaps to give them priority. In theory, the greater threat may be to a species that does not have local value. I personally agree with advocating for taking local and indigenous culture into account when making conservation decisions, but that is a separate idea to the prediction of threats. Perhaps reorganize into separate paragraphs or reword if I am misunderstanding what was written.

P10 L310- A species may overlap with multiple cultures. Even if a downward trend in a species with local importance where acknowledged, work to conserve the species might be focused outside of its area of cultural importance. Acknowledge there may be a spatial mismatch the people who find the species important and the population that is in decline.

P10 L3210-I was unsure what “enhancing the social bases needed for conservation action” means exactly. Reword.

P11 340-Section ‘How many species have been ‘saved’ from extinction?’. The paper’s flow seemed to go from push factors, to push back factors, to changes that can be made to account for these factors and then suddenly ends on a section on the specific effects push back factors have had in preventing extinction. It was not clear why this section was not earlier in the paper after or integrated into the section on push back factors. Consider either restructuring or making it clearer what the logic behind the current structure is.

P12 L371- Fix typo “U.”

P12 L372-Add a period between “biodiversity” and “Most”

P12 L383-Do not capitalize the “T” in “Threatened”

---

## [Editor Report]

Thank you for this contribution. I enjoyed reading this manuscript and agree with both reviewers that this could make a worthwhile addition to the literature on extinction, by highlighting what is certainly an under-studied and under-appreciated aspect of the extinction process. The reviewers have a list of major and minor suggestions for improvement, and I think it will be important to try and attend to most of these to provide a complete and well balanced review paper. Some key points from the reviews and my own reading of the manuscript are as follows:

I’m not sure I agree entirely with R1 that the introduction is long-winded – I think it does quite a good job of setting the scene. You could consider streamlining the most basic of the background material to get to the point a bit quicker, but I think its OK.

The lack of discussion of Chinese (and other) traditional medicinal uses of wildlife seems to be an obvious gap – surely this is an important driver of some of the biocultural mechanisms you discuss, and warrants more discussion?

At line 67 you say that extinction is an explicitly biocultural phenomenon, but I’m not sure that I agree with this. I’d argue that as a process, extinction is biological, and the biocultural mechanisms are part of the many different ways that anthropogenic environmental change drives the process, but are not part of the process itself. Extinction can (and does) still happen in the absence of conscious human awareness that a species is going extinct, of taxonomic decisions, or even awareness that it exists. But I’m happy to be convinced otherwise!

The concept of prioritization in making conservation decisions is implicit throughout your manuscript, but is never discussed explicitly. Given the growing trend for systematic, structured, and algorithmic decision making, it might be worth thinking about how you could integrate this with biocultural aspects. This could be relevant, for example, for your suggestions about incorporating biocultural measures into the Red List, which (as it stands) are a bit vague and under-developed. The great strength of the Red List is the simplicity, objectivity, and generality of its criteria. I think any proposals to value-add to the Red List should be prepared to live up to these standards.

---

## [Reviewer Report]

I am satisfied with the changes made and responses to my comments. I think this is an interesting and worthwhile paper.

---

## [Reviewer Report]

Dear authors,

I have enjoyed reading this new version of the manuscript. All the points that I raised in the first revision have been corrected and integrated very well into the text. I would say that the manuscript is now ready for publication.

I have reflection about the opportunity to differentiate “Indigenous Peoples and local communities”. I understand perfectly the point behind this differentiation, but this difference is not always present (in Europe, for example, the two groups are often perfectly overlapping) and could be a starting point for reflection to improve our approach to the perception of nature.

---

## [Editor Report]

Thank you for your revised manuscript. Both the two reviewers and myself are happy that you have adequately addressed their original concerns.